# Life-On-Hold: Lanthanoids Rapidly Induce a Reversible Ametabolic State in Mammalian Cells

**DOI:** 10.3390/biology10070607

**Published:** 2021-06-30

**Authors:** Anastasia Subbot, Sabina Kondratieva, Ivan Novikov, Natalia Gogoleva, Olga Kozlova, Igor Chebotar, Guzel Gazizova, Anastasia Ryabova, Maria Vorontsova, Takahiro Kikawada, Elena Shagimardanova, Oleg Gusev

**Affiliations:** 1Research Institute of Eye Diseases, 119021 Moscow, Russia; kletkagb@gmail.com (A.S.); i.novikov@niigb.ru (I.N.); 2Institute of Fundamental Medicine and Biology, Kazan Federal University, 420111 Kazan, Russia; sabinakondr@gmail.com (S.K.); olga-sphinx@yandex.ru (O.K.); grgazizova@gmail.com (G.G.); rjuka@mail.ru (E.S.); 3Prokhorov General Physics Institute of the Russian Academy of Sciences, 119991 Moscow, Russia; nastya.ryabova@gmail.com; 4Kazan Science Centre, Kazan Institute of Biochemistry and Biophysics, Russian Academy of Sciences, 420111 Kazan, Russia; negogoleva@gmail.com; 5Laboratory of Molecular Microbiology, Pirogov Russian National Research Medical University, 117997 Moscow, Russia; nizarnn@yandex.ru; 6Institute for Regenerative Medicine, Lomonosov State University, 119991 Moscow, Russia; maria.v.vorontsova@mail.ru; 7Endocrinology Research Center, 115478 Moscow, Russia; 8Laboratory of Orphan Diseases, Moscow Institute of Physics and Technology, 141701 Moscow, Russia; 9Anhydrobiosis Research Group, Division of Biomaterial Science, Institute of Agrobiological Sciences, National Agriculture and Food Research Organization, Tsukuba 305-8634, Japan; kikawada@affrc.go.jp; 10Department of Regulatory Transcriptomics for Medical Genetic Diagnostics, Graduate Schoole of Medical Sciences, Juntendo University, Tokyo 113-8421, Japan; 11RIKEN Center for Integrative Medical Sciences, RIKEN, Yokohama 351-0198, Japan

**Keywords:** cells metabolism, anabiosis, ametabolic state

## Abstract

**Simple Summary:**

We found that incubation with a solution containing ~50 mM neodymium (one of the rare-earth elements, REE) induces a rapid of active metabolism in mammalian cells. We have named this state REEbernation and found that the process involves a rapid replacement of calcium with neodymium in membranes and organelles of a cell, allowing it to maintain its shape and membrane integrity under extreme conditions, including vacuum. Furthermore, phosphate exchange is blocked because of non-dissolvable neodymium salts formation, which “discharged” the cell. We also showed that REEbernation is characterized by instant shutting down RNA transcriptional activity in the cells, providing an intriguing opportunity to study a snapshot of gene expression at a given time point. Finally, we found that the REEbernation state is reversible, and we could restore the metabolism and proliferation capacity of the cells. The REEbernation provides a new method to reversibly place a cell into “on-hold” mode, opening opportunities to develop protocols for biological samples fixation with a minimum effect on the omics profile for biomedical needs.

**Abstract:**

Until now, the ability to reversibly halt cellular processes has been limited to cryopreservation and several forms of anabiosis observed in living organisms. In this paper we show that incubation of living cells with a solution containing ~50 mM neodymium induces a rapid shutdown of intracellular organelle movement and all other evidence of active metabolism. We have named this state REEbernation (derived from the terms REE (rare earth elements) and hibernation) and found that the process involves a rapid replacement of calcium with neodymium in membranes and organelles of a cell, allowing it to maintain its shape and membrane integrity under extreme conditions, such as low pressure. Furthermore, phosphate exchange is blocked as a result of non-dissolvable neodymium salts formation, which “discharged” the cell. We further showed that REEbernation is characterized by an immediate cessation of transcriptional activity in observed cells, providing an intriguing opportunity to study a snapshot of gene expression at a given time point. Finally, we found that the REEbernation state is reversible, and we could restore the metabolism and proliferation capacity of the cells. The REEbernation, in addition to being an attractive model to further investigate the basic mechanisms of cell metabolism control, also provides a new method to reversibly place a cell into “on-hold” mode, opening opportunities to develop protocols for biological samples fixation with a minimum effect on the omics profile for biomedical needs.

## 1. Introduction

An active metabolism is one of the basic signs of life. At the same time, there are several ways in which living cells can be preserved in a non-metabolic state. Inspired by many examples of organisms surviving after freezing in natural circumstances, where cellular kinetics are brought to a halt by a drop of temperature and in the presence of cryoprotectants, cryopreservation is actively used in biomedical and biotechnological sciences for long-term preservation of cells, tissues, and simple organisms [1].

A naturally occurring example of ametabolic preservation is anhydrobiosis, a process, where, under ambient temperature conditions, a drastic decrease in water content inside a cell occurs, followed by an increase in intracellular viscosity (in many cases associated with the replacement of water by the disaccharide trehalose), concluded by termination of metabolic activity. Anhydrobiosis, which has evolved independently in several taxa, typically requires a prolonged drying phase during which drastic changes in both gene expression and biochemical/biophysical conditions in the cells take place [2]. Recovery of cells from both cryopreservation and anhydrobiosis (for example in tardigrades and anhydrobiotic midge) is also associated with marked changes in both genetic and biochemical/biophysical profiles [3,4,5].

Rare earth elements (REEs) form a group comprised of yttrium, scandium, and the 15 lanthanides (lanthanum, cerium, praseodymium, neodymium, promethium, samarium, europium, gadolinium, terbium, dysprosium, holmium, erbium, thulium, ytterbium and lutetium). In general, lanthanoids are characterized by extremely low availability in the biosphere and do not form part of biological molecules; one known exception is the bacterium *Methylacidiphilum fumariolicum*, where lanthanoids act as a co-factor for methanol dehydrogenase [6]. On the other hand, the lanthanoids exhibit low toxicity and are considered to have potential as anti-cancer agents due to their effect on the proliferation of cancer cell lines [7] in addition to their capacity to block virus replication [8]. The lanthanoids are also actively used in microscopy as contrasting agents since they bind to structures responsible for cellular metabolism and architecture. We recently reported a new method of whole-cell sample preparation for scanning electron microscopy (SEM) based on staining/fixation with a rare earth metal alone [9].

In the current paper, we analyze the effect of neodymium on a mammalian cell line and find that it instantly halts all detectable metabolic processes, including transcription. This “life-on-hold” state can be maintained for at least a few hours and later reversed, so that normal cellular activity, including proliferation, is resumed.

## 2. Materials and Methods

### 2.1. General Study Design

In the current study, we analyzed several regimes of incubation of the cell culture with isotonic lanthanoid-containing solution. Figure 1 represents a general scheme of sampling and conducted type of analysis.

### 2.2. Cell Culture

Mammalian cell line A-549 was used (immortalized human bronchopulmonary epithelium). The cell line was kindly provided by the Laboratory of Experimental Immunology and Virology, NRC, Children’s Health, Ministry of Health of Russia. A-549 cells were grown in RPMI-1640 medium with glutamine and 10% fetal bovine serum (HyClone, Logan, UT, USA) at 37 °C, 5% CO_2_, and 100% humidity. Growth media, supplements, and washing solutions were by Gibco (Bleiswijk, The Netherlands).

Cells were seeded as follows:(1)For the transcriptome study: on 3.5 cm diameter Petri dishes (SPL Life Sciences, Pocheon-si, Korea) at a density of 60 × 10^3^/cm^2^ and grown to a density of 300 × 10^3^/cm^2^ for three days before harvesting;(2)For proliferation assay: on 12-well plate (SPL life sciences) at a density of 40 × 10^3^/cm^2^ one day before the experiment;(3)For SEM: histochemical staining on the 3.5 cm diameter Petri dishes at a density of 40 × 10^3^/cm^2^ one day before the experiment;(4)For confocal microscopy: on the POC-R2 (PeCon GmbH, Erbach, Germany) closed perfusion cell cultivation system at a density of 100 × 10^3^/cm^2^ one day before the experiment.

### 2.3. Lanthanoid Treatment

An isotonic lanthanoid-containing solution BioREE (Glaucon, Moscow, Russian Federation), comprised of 50 mM neodymium chloride and 6 mM Magnesium chloride as stabilizer agent, was used in our work for lanthanoid treatment. Previously, this reagent was used as a contrast agent to prepare objects for SEM.

Before exposure to the BioREE solution, cells were washed with isotonic NaCl (to remove any growth medium components).

Cells were kept in BioREE at 21 °C and atmospheric environment for the following time periods:20 min for confocal and electron microscopy1 h and 4 h for proliferation assays20 min and 4 h for transcriptome study20 min, 1 h, and 4 h for histochemical study

### 2.4. Treatment to Replenish the Ions Concentration

Part of the probes was exposed after lanthanoid treatment with the following solutions sequentially:(1)0.9% NaCl solution—wash out remaining REE ions;(2)PBS, pH 7.4—returning Pi;(3)HBSS with Ca^2+^ and Mg^2+^ (the concentration of Ca^2+^ in this solution was increased to 3.9 mM)—returning calcium ions;(4)0.9% NaCl solution—wash out of excess calcium;(5)Growth medium (RPMI-1640 with 10% FBS).

In a time-lapse confocal microscopy study, the cell medium was replaced by Bio-REE solutions and then returned back directly during the microscope session.

### 2.5. Control Groups

Control cell samples were incubated in isotonic NaCl. Relative toxicant control specimens were incubated in 20% ethanol. Dead control specimens were incubated in 40% ethanol instead of BioREE.

### 2.6. Histochemical Staining

After experimental treatment, the cells were incubated 5 min at RT with 500 nM MitoTracker Orange CMTMRos (Molecular Probes, OR, USA) for analysis of mitochondria state and with 500 nM DAPI (Invitrogen, Carlsbad, MA, USA) for membrane integrity assay. After PBS washing, the cell culture was visualized under an epifluorescence microscope.

To visualize mitochondria cells under a confocal laser scanning microscope, the cells were incubated with 100 nM MitoTracker Orange CMTMRos for 20 min in growth medium at 37 °C in 5% CO_2_ before exposure to experimental solutions.

Membrane integrity and relative number of viable cells and stained (dead) cells were calculated also in the Goryaev chamber after detachment of cells with trypsin/EDTA solution (0.25%) and mixing in equal volumes with trypan blue solution 0.4% immediately after treatment for ions concentration replenishment and after 24 h of restoration in growth medium. The data were acquired in three independent series of observations with the total number of replicates being 13 for Nd 1 h, 7 for Nd 4 h, and 12 for control.

### 2.7. Electron Microscopy

After lanthanoid treatment, cells were washed with distilled water and analyzed using a scanning electron microscope (EVO LS10; Zeiss, Germany) equipped with a LaB_6_ cathode in extended pressure mode at 70 Pa with 21–27 kV acceleration and 360–520 pA current. Images were captured in the BSD (back-scattered electrons detection) mode.

### 2.8. Epifluorescence Microscopy

Cells were visualized at AxioVert A1 FL-LED with filter set 01 for DAPI (EX BP 365/12, BS FT 395, EM LP 397) and filter set 15 for MitoTracker (EX BP 546/12, BS FT 580, EM LP 590). Photos were taken on a Canon 600D camera.

### 2.9. Confocal Microscopy

To acquire images, a laser scanning microscope (LSM-710; Zeiss, Germany) with a 63× oil Plan-Apochromat NA 1.4 objective was used.

MitoTracker fluorescence was recorded at a wavelength range of 570–660 nm when excited by laser radiation with a wavelength of 561 nm (0.1–2% laser power). An overlay of the fluorescent MitoTracker image (red pseudo-color in images) and the transmitted light image (grey color) was obtained. Images were obtained with the following acquisition parameters: image scanning frame 2048 × 2028 pixels with a speed of 0.39 µs/pixel and a waiting time between frames of 0. The scanning time of each image was 1.636 s with zoom = 1 and frame size = 134.89 µm (0.066 µm/pixel).

### 2.10. Analysis of Cell Motion

To estimate the amount of cell movement during lanthanoid treatment, a mathematical implementation was used. Cell boundaries were recognized on time series images obtained with the LSM-710 microscope using the algorithm described previously [10]. We further estimated the speed of movement of cell borders in frames using the method for finding the optical flow pattern [11].

The whole algorithm is shown in the Appendix A.

### 2.11. Proliferation Assays

Cell population growth dynamic was evaluated by measuring the area occupied by cells (analysis of confluence) for 5 days after re-activation using an IncuCyte S3 cellular analyzer (Essen BioScience, Ann Arbor, MI, USA).

Statistical processing and construction of growth curves were performed using the IncuCyte Zoom software environment.

### 2.12. Transcriptome Experiments

To analyze gene expression response, the specimens were washed with isotonic NaCl solution after lanthanoid treatment, and either their RNA was immediately extracted for RNA-seq or cells were treated for ions concentration replenishment and placed into an incubator in a growth medium (37 °C, 5% CO_2_, and 100% humidity) for 60 min to start recovering their metabolic activity before subjecting them to RNA-seq profiling.

An additional set of cells was subjected to the classical protocol including non-freezing preservation with RNAlater solution (Invitrogen) followed by harvesting for RNA after 20 min. All experiments were done in triplicates to ensure statistical significance.

#### 2.12.1. RNA Extraction and Sequencing

RNA extraction was done with a RNeasy Mini Kit (Qiagen, Hilden, Germany) according to the manufacturer’s instructions and stored at −80 °C until use. DNA residues were eliminated by DNase I treatment using a DNA-free kit (Ambion, Austin, TX, USA). Enrichment of mRNA was performed using a NEBNext Poly(A) mRNA Magnetic Isolation Module (New England Biolabs). Further, cDNA libraries were prepared with a NEBNext Ultra II Directional RNA Library Prep Kit (New England Biolabs). Qualitative and quantitative evaluation of the RNA and cDNA libraries was performed with a Qubit fluorometer (Invitrogen, USA) and 2100 BioAnalyzer (Agilent Technologies, Santa Clara, CA, USA) using the Agilent RNA 6000 Pico Kit (Agilent Technologies). Sequencing was performed on the HiSeq2500 platform (Illumina, San Diego, CA, USA) using single read sequencing at the Joint KFU–Riken Laboratory, Kazan Federal University (Kazan, Russia).

#### 2.12.2. Data Analysis

RNA reads (60 bases long, 10 samples in total, two replicates for each experiment) were mapped to the human genome (GRCh38) using the hisat2 tool (version 2.1.0). The number of reads overlapping human genes (Gencode annotation, version 32) was calculated with htseq-count (version 0.6.0). Further analysis was conducted in R (version 3.6.1). Differential gene expression analysis was based on the GLM approach using the edgeR package (version 3.28.0). To reduce noise (by low-signal filtration), only genes having at least 10 reads in at least six samples were chosen for this analysis. A gene was considered to be differentially expressed if its Benjamini–Hochberg-corrected *p*-value was less than 0.05. When fitting the model, the largest of three biological coefficients of variation (common, trended, tagwise) was used as a dispersion.

## 3. Results

### 3.1. General Morphology of the Cell, Cell Movement Analysis

As we demonstrated before, the lanthanoid treatment stabilizes cells integrity and acts as a contrast agent for scanning electron microscopy (SEM) in BSD mode (Figure 2). Indeed, both external and internal composition of cells could be clearly visualized, which suggests pervasion of Nd into the interior of the cells with an integration into multiple cellular organelles and membranes. We also noted that cells preserved their 3D shape under vacuum conditions, which suggests a powerful inhibition of membrane channels [9].

In preliminary observations, we noted the lack of organelles movement and decided to study this fact in more detail by observing the dynamics of the lanthanoid treatment process via confocal microscopy time series (Appendix A). The overall amount of cell movement decreased to Brownian level in 10 min of exposition start. Upon recovery of the cells (washing and restoration of normal growth medium), cell movement rapidly increased until it exceeded the original level (Figure 3A). This phenomenon was not observed in the study of control samples (Figure 3B, Appendix A).

We did not observe any alteration of cell morphology after lanthanoid treatment. There were no gross disturbances of cell structures; the shape of the cells remained intact during exposition time; and right after washing, fluorescent labelling of mitochondria could be maintained throughout the procedure, which confirms the preservation of the mitochondrial membrane integrity (Figure 4).

### 3.2. Preservation of Organelles in Cells after Recovery from Lanthanoid-Induced State

The number of cells with damaged membrane (and thus permeable for trypan blue) was 7.86 ± 0.36 after 1 h NdCl_3_ treatment and replenishing the ions concentration (BR1hRev group). After 24 h of restoration, this value was 5.22 ± 0.55.

For cells treated with NdCl_3_ for 4 h, this index significantly reduced after 24 h of restoration (from 53.1 ± 2.13% to 28.67 ± 1.95%), indicating that cells had recovered.

The number of trypan-positive cells in control vessels was unchanged and amounted to 1.88 ± 0.28 at 0 h and 1.22 ± 0.21 at 24 h.

We performed a control series with ethanol treatment (as known toxicant) and mitotracker staining to illustrate the presence of a «live pattern» of distribution after lanthanoid treatment (Appendix A).

We also obtained results on the permeability for DAPI dye. Data are shown in comparison with control, dead control, and relative toxicant control.

It can be concluded that the effect of lanthanoid treatment does not lead to an increase in mortality as indicated by an increase in membrane permeability and does not change the pattern of mito-tracker accumulation for up to 4 h.

### 3.3. Proliferation of Cells after Lanthanoid Treatment

Knowing that a short treatment with lanthanoids does not lead to visible damage or major cytotoxic consequences for the cells, we further analyzed whether it would have a long-lasting effect on cell biology. We re-initiated A-549 cells to normal activity levels after lanthanoid treatment and found that the part of cells could successfully restore proliferation after four hours in the REEbernation state (Figure 5B).

Within the first 24 h after re-activation of the cells incubated for 4 h with lanthanoids, the space occupied by the cells increased from 19.5% to 25.5%. Remarkably, the main increase of the cells was observed during the first 180 min after reactivation of the cells. After next 48 h, there was an active growth of the cells, followed by returning of the usual cell dynamics around 72 h after reactivation from lanthanoids-induced state.

Please note that in the analysis of the space occupied by the cells, dead cells were also counted, if they adhered to the plastics. Such cells would mask the proliferation during the first 24–48 h after reactivation. This is why there is an initial “plateau” on the graphs, showing the dynamic of cell growth.

The shorter (in time) lanthanoid treatment did not change the post-reactivation dynamic of normal growth of cell populations (Figure 5A).

### 3.4. Differential Gene Expression in Cells in Return to Lanthanoid Treatment

To estimate the effect of REEbernation on the genetic machinery of the cell, we analyzed changes in gene expression in the cells subjected to 20 min (BR20 min) and in those subjected to 4 h (BR4 h) of lanthanoid treatment, and then 60 min after restoring cells to normal activity in growth medium (BR4 hRev). Intriguingly, there were very few changes in gene expression in cells after 20 min treatment, compared to the control group (Figure 6A). Thus, despite ambient temperature, the neodymium treatment resulted in almost immediate termination of gene expression, and with no degradation of RNA (Appendix A), which would mean that cells are likely to represent a snapshot of mRNA expression at the moment of exposure to BioREE. In contrast, cells prepared with RNAlater solution showed marked changes in their mRNA expression profile (Figure 6E), which is likely a consequence of the toxic effect by the fixation solution. The aforementioned results are in agreement with previous reports by other research groups [3,12].

Restoring the cells to normal growth medium after lanthanoid treatment (sample BR4 hREV) was associated with a remarkable switch in gene expression (Figure 6C,D), which illustrates restoration of transcriptomic machinery activity.

## 4. Discussion

The possibility of “pausing” cell activity with the option to resume all metabolic processes including proliferation after the hibernating agent has been removed would open up exciting new opportunities for short- and long-term preservation technologies for biomedical and other applications. While true anhydrobiotic organisms can survive without water and effectively preserve their biomolecules, the induction of such a state requires an extended initiation of special molecular mechanisms associated with massive synthesis of protective substances, thereby gradually changing the cell morphology and metabolite composition, as well as the omics profile [13,14,15]. While anhydrobiosis is an attractive model for biomimetic studies, we are still far from developing an effective protocol for preserving living mammalian cells at ambient temperature in the dry form, or even in “paused” mode, via adoption of the biomolecular strategy used by anhydrobiotic organisms [16]. Another issue is that there is still a large difference between the natural state of mammalian cells we might wish to preserve and those cell types capable of anhydrobiosis or those stored cryogenically with protectants, for the latter two categories of cells are characterized by adaptive changes (in the case of anhydrobiosis) or responses to cryoprotectants and freezing itself. In contrast, we found that neodymium treatment and the accompanying immediate shutdown of cellular metabolism results in a negligible negative effect on the transcriptome of the treated cells. Thus, RNA transcription is rapidly blocked by NdCl_3_ treatment (Figure 5).

### 4.1. Nature of Freezing Processes

We assume that the phenomenon we see is direct evidence that at ambient temperature (room temperature, RT) it is possible to chemically imitate some of the processes which are typical of reversible cell freezing. The main consequence of freezing is a rapid cessation of all chemical reactions in the cells. This is achieved by two mechanisms. First, the lowering of temperature itself limits the speed of most reactions. Second, the phase transition of intracellular fluid to a solid state leads to the fact that molecules cannot be involved in a chemical reaction, since they cannot overcome the physical distance between themselves and their partner reactants without a mobile carrier [17]. In a frozen cell, this process blocks even those chemical reactions that may still be thermodynamically possible. Even non-specific oxidation of organic substances cannot be supported, since it is limited by diffusion rate of defects and inclusions in an amorphous vitrified electrolyte and solid phase transition products. The speed of this process is comparable to that of self-diffusion in ice crystals, salts, and clathrates [18]. Thus, freezing does not affect the structure and relative position of chemical molecules in the cell, but it dramatically slows all chemical processes and the flow of biological time stops. Cryopreservation provides the cell with chemical and structural inalterability, which makes the process reversible. Therefore, it is only necessary to restore the properties of the medium, i.e., transfer the intracellular content to a liquid state by raising the temperature, to return the cell to its original condition [19].

In our study we were able to partially mimic the chemical aspects of a frozen cell state, but at room temperature, using a metal from the lanthanum group.

### 4.2. Presupposition

The trivalent cation of neodymium has three features that make it well suited for simultaneous reversible shutdown of a wide range of metabolic processes in the animal cell:The electrochemical activity of lanthanoid cations and their effective ionic radius allow them to affect all reactions involving calcium in the cell, as well as to replace calcium in organometallic compounds.It has a high reactivity with bases of weak and medium polybasic inorganic acids. The salts arising in the course of such reactions are insoluble and biologically inert in the intracellular environment.The lanthanoid cation and the paired chloride anion can exist in balanced solution under physicochemical conditions corresponding to the normal environment of the cell interior.

Stable solution of lanthanum chloride (or any cation from this group) by itself has low pH. Here, the meta-stable conditions under neutral pH can be obtained by addition of magnesium to the system.

Consequently, a metastable solution based on LnCl_3_ can be created with pH, osmolarity, and temperature parameters that correspond to the environmental conditions that are vital for most cells.

The first two qualities of lanthanoids determine their suitability for quickly switching off the Ca^2+^, P_i_, and CO_2_ exchange chains, and the third quality determines the technological feasibility of delivering REE cations to the cell without significant chemical damage.

At first approximation, all elements of the lanthanum group exhibit very similar chemical properties. In this study, we have demonstrated the effectiveness of only one metal from this group, neodymium, although in preliminary tests we also used samarium, dysprosium, and lanthanum, which gave similar effects (unpublished data). Neodymium gave the best results in our recent electron microscopy study, where we tested several lanthanoids to determine the best supravital agent for contrasting the internal structures of the cell [9]. This may be because neodymium is better at entering the cell, but in the absence of quantitative biophysical measurements, choosing one among these chemically very close metals of this group is somewhat arbitrary at this stage. Our study, in this regard, is in agreement with previous reports where the influence of lanthanoid cations on various processes in the cell are described [8,20,21].

### 4.3. Blockage of Calcium Metabolism—“High Temperature Freezing”

A large part of intracellular metabolism is calcium-dependent, and one major component of the observed effect of neodymium is inhibition of Ca^2+^ exchange. While such inhibition is usually explained by the principle of electrochemical blockade of the ion channel, we assume that at least some proportion of the effect relates to isostructural chemical exchange reactions during ion transport, as shown previously with purified proteins and later in vitro for myocytes [22]. The ideal substitution of the transported pair of cations can be described as:2Ca^2+^ → Na^+^ + Nd^3+^2Ca^2+^ → K^+^ + Nd^3+^

Which pertains to the Ca-dependent ATPase, for example [23]. This mechanism was earlier confirmed by observing the different levels of toxicity of lanthanoids compared to K^+^ and Na^+^ on invertebrates and mammal/plant cells [24,25,26]. Furthermore, treatment with lanthanoids extends the viability of Arabidopsis under conditions of potassium deprivation and generally prolongs the vase life of cut flowers [27,28]. Thus, shutdown of the Ca^2+^ pump suppresses storage of energy in the macroergic bonds of nucleotide phosphates.

The blockade of voltage-activated calcium channels would follow a different scenario, but also seems to occur as shown by verapamil blocking the entry of lanthanum into the cell [29]. Turning off this system cancels the reactions triggered by Ca^2+^-dependent currents. It is also known that lanthanoids at concentrations higher than 0.25 mM block Na^+^/Ca^2+^ channels, with some evidence that this is reversible [30]. The effect of lanthanoids we describe in this report is also, perhaps surprisingly, reversible (Figure 3 and Figure 4). Our data suggest that high concentrations of lanthanoid completely impede the exchange of bound calcium in the cell, which resembles the effects of freezing. However, in the case of neodymium, the reason for calcium exchange being blocked is not the presence of a physical barrier (i.e., ice formation), but the shutdown of calcium-dependent metabolism.

The timespan from the start of exposure of a cell to neodymium to the halt of key cellular processes is likely to be very short. This is confirmed by the minimal cell response at the transcriptional level (Figure 5). The number of “responding genes”, with fold change more than 2, was 0 genes for 20 min exposition and only 2 genes for 4 h exposition, which is extraordinary for more conventional methods of RNA conservation (for RNA-later preparation this index was 61) (See Appendix A).

### 4.4. Conversion of Free Phosphate into an Insoluble form–“Discharging”

The thermodynamic parameters of lanthanoid ions allow them to form stable compounds with phosphate. For neodymium, the equilibrium parameters for the formation of NdPO_4_ were described for a closed system involving the cytoplasm electrolytes [31]. The energy balance for the binding of lanthanoids to P_i_, which is in the form of HPO_4_^2−^ and H_2_PO_4_^−^ ions in an active cell, is sufficient for a rapid progress of this reaction and the maintenance of its product as a biologically inert simple phosphate, lanthanoid (III), orthophosphate [31].

The solubility of lanthanoid phosphates, although slightly higher than that of calcium and magnesium phosphates, almost eliminates those anions from the solution due to the formation of a solid phase. All lanthanoid phosphates have a low solubility of -log Ks: Eu 12.2; La 11.0; Ce 11.35; Sm 12.1; Pr 11.6; Nd 11.8; Gd 12.2; Tb 12.4; Dy 12.5; Yb 12.9; Y 12.6. This results in very limited substitution of the cation in the solid phase such that there is little release of P_i_ and consequently poor availability of P_i_ for metabolic processes. The lack of free P_i_ drastically reduces cytoskeletal movements and significantly limits transport of the essential components of chemical reactions [32].

A particular consequence of lanthanoids presence in the cell with respect to phosphate metabolism is the promotion of rapid and non-enzymatic conversion of all ATP molecules to cAMP by lanthanoids [33]. Reverse phosphorylation to restore local ATP/ADP/AMP balances is not possible due to the deficiency of free P_i_ in the cell. This deprives the cell of access to an energy supply, preventing its activity.

### 4.5. Reversibility of REEbernation State

Similar to the frozen state, the chemical phenomena described above are not lethal to the cell for a certain period and do not fatally compromise the molecular composition of organic macromolecular compounds. Although lanthanoids are capable of breaking some bonds based on the phosphoric acid residue, this process does not seem to proceed rapidly, and for a long time, the chemical composition of the cell remains unchanged, except for the chemical events described above. There is evidence that Nd^3+^ treatment leads to a relatively slow degradation of RNA compared to other lanthanoids [34]. Our results suggest that the cell lacking ATP and P_i_, and with inhibited Ca^2+^-dependent metabolic systems but with intact structural and molecular architectonics maintains the ability to fully recover and proliferate. Reversibility is achieved by returning the cell to a state where there are sufficient molecules of the phosphoric acid residue and an excess of Ca^2+^ to promote normal metabolism. At low lanthanoid concentrations (up to 0.25 mM), the potential reversibility of Ca^2+^ metabolism was reported previously [30]. Our experiments showed that cell structures have sufficient permeability to replenish the intracellular concentration of each of these ions in a concentration gradient. For the “revitalization” of the cells, it was enough to remove neodymium salts with isotonic salt solutions and to restore normal levels of calcium and phosphoric ions using commonly available reagents (see Section 2).

We observed full recovery of the cells, including their ability to proliferate, after at least 4 h of REEbernation state, but we expect, by optimizing recovery conditions, that we can extend this timeline to days and even months. This would make it comparable to and competitive with currently used methods of preservation. Lastly, the biochemical mechanisms of REEbernation are of exceptional interest and their further investigation should deepen our understanding of the basic principles of cellular biology.

## Figures and Tables

**Figure 1 biology-10-00607-f001:**
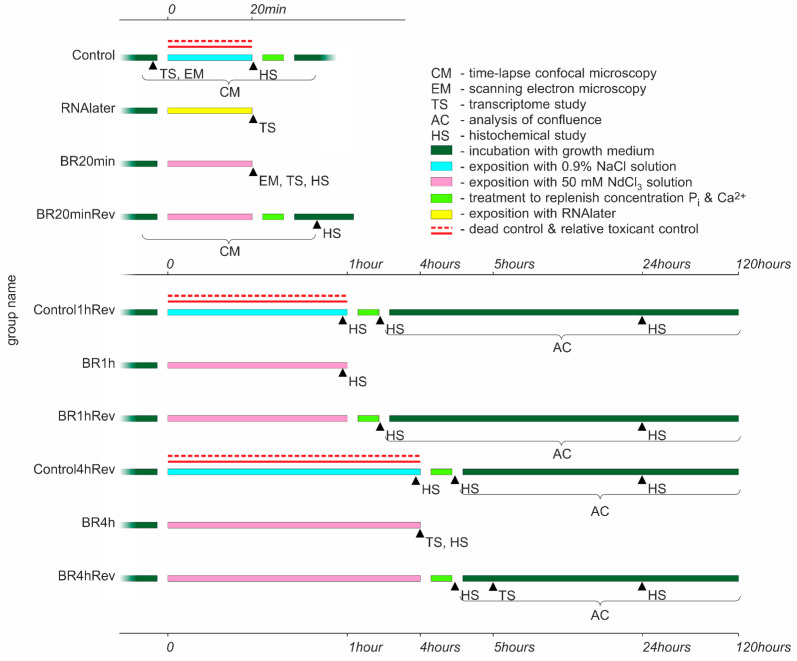
Sampling and type of analysis conducted on the cell culture after incubation with isotonic lanthanoid-containing solution and control substances.

**Figure 2 biology-10-00607-f002:**
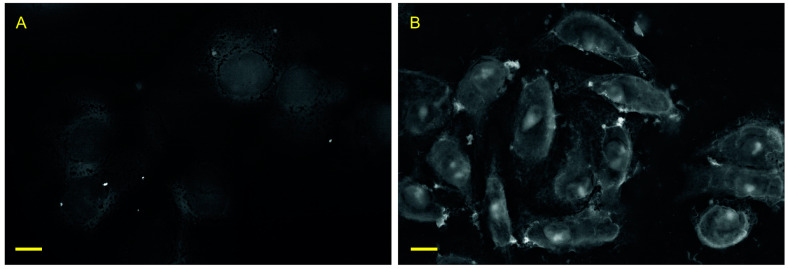
Scanning electron microscopy (SEM) image (in back-scattered electrons detection (BSD) mode) of control cells without treatment (**A**) and BR20min-cells after 20 min of lathanoid treatment (**B**), followed by water washing and 10 min at vacuum (70 Pa). Scale bar: 10 µm.

**Figure 3 biology-10-00607-f003:**
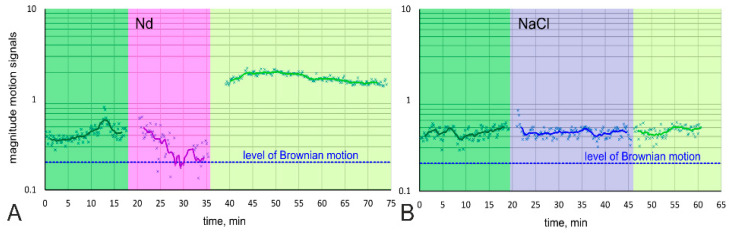
Diagram of changes in the magnitude of cell motility before (dark green), during (magenta), and after (light green) lanthanoid treatment (**A**) and control NaCl treatment (**B**).

**Figure 4 biology-10-00607-f004:**
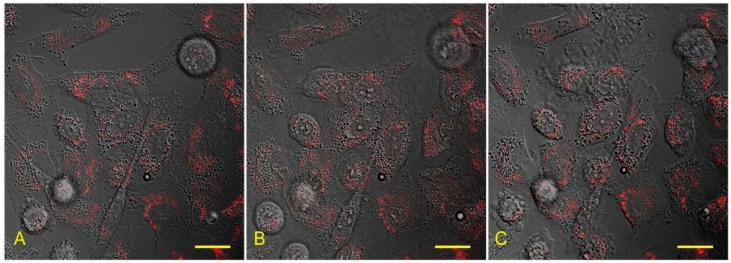
Overlay of the fluorescent (red channel—MitoTracker) and the transmitted light image of the cells before (**A**), during (**B**), and after recovery (**C**) from lanthanoid treatment-induced state. Scale bar: 20 µm.

**Figure 5 biology-10-00607-f005:**
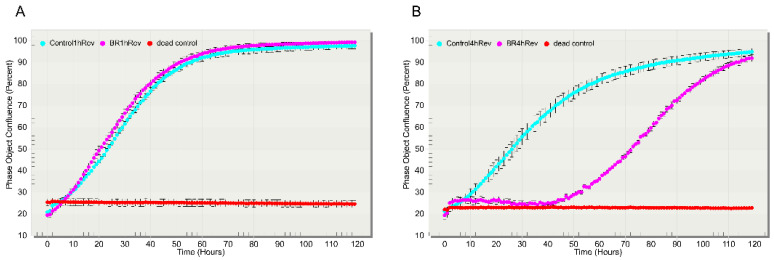
Growth curves depicting the dynamics of cell proliferation after lathanoid treatment (purple line) for 1 h BR1hRev (**A**) and 4 h BR4hRev (**B**) in comparison with control (cyan line) and dead control (red line).

**Figure 6 biology-10-00607-f006:**
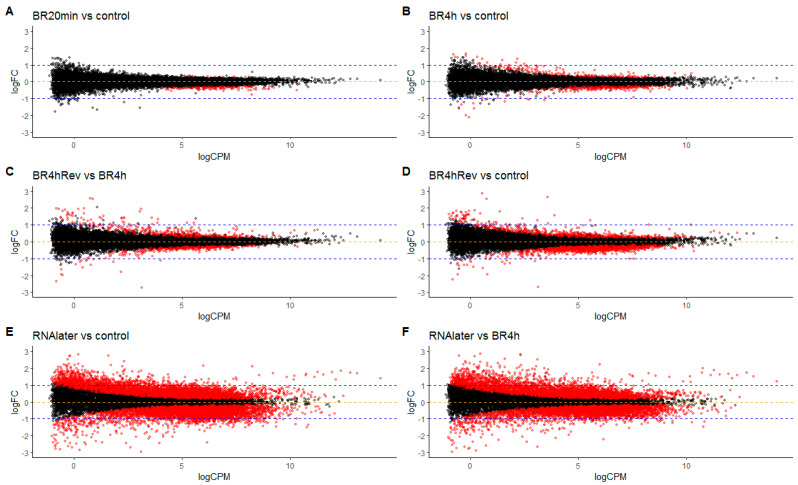
Changes in gene expression in the cells with neodymium (**A**,**B**), RNAlater (**E**,**F**), and after restoration of the cells to growth medium (**C**,**D**). Each point represents a gene with its average level of expression (logCPM, where CPM is “Counts Per Million reads”) and the degree of expression change between given pairs of samples (logFC, where FC is “Fold Change”). The specific logFC rate indicates changes of gene expression in the first sample in pairs compared to the second one. Red points correspond to genes with statistically significant changes of expression.

## Data Availability

Original files with RNA expression raw data are available upon request.

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
