# Peer review of "Life-On-Hold: Lanthanoids Rapidly Induce a Reversible Ametabolic State in Mammalian Cells"

_biology, 2021, doi:10.3390/biology10070607_

Round 1

Reviewer 1 Report

The ability to reversibly halt cellular processes is valuable for many reasons. The results shown here are very intriguing and provide the foundation for future research into the possible widespread use of lanthanides for cellular preservation. With the addition of several important controls and clarifications about what was done experimentally, these experiments should be of great general interest.

In the “graphical abstract”, the illustration of the cell at position “1” has a series of smaller, more transparent illustrations layered behind it. I am guessing that these may be intended to convey that the cell is “alive” and/or “dynamic”. However, I don’t think that this meaning comes across well. I think the green color of that illustration, coupled with the “play button” image, may be sufficient to convey the intended meaning.

What is the difference between “lanthanides” and “lanthanoids”?

What exactly is shown in Fig. 2? The figure legend provides conclusions but not a description of what was done experimentally. If I assume that the image represents SEM of cells following exposure to neodymium, then in order to be able to reach the stated conclusion that cell shape and retention of liquid is preserved under a low vacuum, I would need to see an image captured in the same way of cells prepared in parallel but not exposed to neodymium, and cells prepared in parallel but not exposed to vacuum. These important controls are missing here and must be provided.

I was not able to locate or view the videos, since I could not find a way to access the Supplementary Information. I imagine that these data would be quite valuable.

Similar to the situation for Fig. 2, in Fig. 3 the figure legend provides conclusions but not a description of what was done. I think that the title of the Figure could be “During exposure to NdCl3, cell movement decreases, while after washing it increases above

initial levels”, but the content of the legend itself should describe how cell motility was measured, with how many cells and under which precise conditions, and what the lines on the plot represent (mean? median?) and what the individual points on the plot represent (individual cells? individual parts of a single cell? a composite value for multiple cells within a single field/frame?). The conclusions should be provided in the body text, not the figure legend.

How was Brownian motion assessed? Were inanimate objects (beads, killed cells) measured in a parallel set of experiments? This is an important control to establish what indeed represents Brownian motion.

It is stated (line 201) that “signs of cell death” were not observed. What kinds of observations would have been taken as a sign of cell death? Are the assays employed capable of detecting cell death? The control experiment suggested above (treating cells in a way as to kill them, to measure what true Brownian motion would look like) should also be performed here to show the reader what death would look like, with regard to morphology and MitoTracker signal as assessed by microscopy. I, for one, do not know what MitoTracker signal would look like in a dead cell of this type.

Indeed, for each of these experimental approaches, it would be important to have a control experiment in which cells are treated in a way that should induce irreversible damage/death, so that the reader knows what the effects of such treatment would look like in the assays employed here, and how they compare to what is observed.

In lines 217-218 it is stated that the treated cells “divided with the same efficiency as control and doubled their number after 24 hours.” It would be important to see these data as values or as a plot. What does “efficiency” mean, in this context? That the same number of cells divided after 24 hours? Or that the fraction of cells that divided did so at the same rate (speed)? Were the measurements that led to this conclusion performed only at a 24-hr timepoint? This is how I interpret the description of the experiment in the Methods section (line 139). If so, additional timepoints in between 0 and 24 hours would be necessary to make conclusions about the “speed” of proliferation, which is an interesting question.

In section 3.3, it is stated that “The number of cells with damaged membrane (and thus permeable for trypan blue) was significantly reduced” and the values of 53% and 29% are given. To help the reader assess the significance of these values, please provide the raw numbers of cells used to calculate these percentages, and whether these experiments were performed with technical and/or biological replicates. Also, how does 53% of cells with damaged membranes compare to the effects of mock treatments (NaCl) or treatments intended to kill cells? Does having 53% of cells trypan-blue-positive represent massive damage, or is this kind of value consistent with other kinds of treatments that may induce some damage that cells are easily able to recover from? The same question applies to 29% trypan-blue-positive: is this value indicative of persistent damage? What percentage of trypan-blue-positive cells would one expect for an untreated or mock-treated culture?

The information provided in Fig. 1 is really only important for interpreting the data in Fig. 5, since the other figures don’t use the same abbreviations to refer to the different treatment and recovery regimens. I therefore suggest that the information in Fig. 1 be moved to a part of Fig. 5, so that the reader can look at them while examining the changes in gene expression.

In Fig. 5, is fold change calculated for the first dataset listed compared to the second one? For example, does a logFC of 3 for “BR4hRev vs control” mean that a particular gene was expressed at much higher levels in the BR4hRev sample compared to the control sample? I suspect that this is the case, but I suggest that the authors include some wording to make that point very clear. This is an important point because the authors say that the changes illustrate “restoration of transcriptomic machinery activity”. If the major effect is a resumption of new transcription, then one might expect the majority of genes to have positive logFC values, but that does not appear to be the case here. In general, I think the authors should comment on the relative direction of the changes and how they interpret this. In the Discussion, the authors say that “RNA transcription is rapidly blocked by

NdCl3 treatment”. However, if there are few changes in transcript levels, then mRNA degradation is also blocked, which is worth noting. Finally, the changes in expression profile for RNAlater are stated as being “marked” and the data in the table in the SI supposedly supports this statement, but it would be extremely valuable to have those RNAlater data also shown in Fig.5 in the same kind of format (scatter plots), for comparison.

The Discussion is rather long and speculative, but I don’t mind this, since I think that what is discussed is relevant and valuable. However, it may provide a better basis for discussion if the authors were to perform some kind of functional analysis of the changes in gene expression that they observe (GO term analysis would be the most obvious). There may be real clues available here that could inform their speculation about the effects of the treatments.

Author Response

The authors would like to sincerely thank the Reviewer for the overall positive assessment of the work and valuable comments that helped us to improve the article.

Reviewer:

In the “graphical abstract”, the illustration of the cell at position “1” has a series of smaller, more transparent illustrations layered behind it. I am guessing that these may be intended to convey that the cell is “alive” and/or “dynamic”. However, I don’t think that this meaning comes across well. I think the green color of that illustration, coupled with the “play button” image, may be sufficient to convey the intended meaning.

Response:

We thank the reviewer for pointing the visibility issue. One of the remarkable effects we tried to illustrate is the fast deferred movement of the cells on the surface of the plastic after re-activation from lanthanoid-induced anabiosis (Fig. 3B). We ask the reviewer to kindly allow us to keep the current scheme of the graphic abstract. At the same time, we will further try to improve the easy-to-catch aspect of the graphical representation of this image and will be very happy to know additional suggestions.

Reviewer

What is the difference between “lanthanides” and “lanthanoids”?

Response

These worlds usually are used synonymously. But we mean here that Lanthanides are elements from lanthanium group,  while lanthanoids mean Lanthanides together with yttrium and scandium. To avoid confusion, we unified the utilization of the word “lanthanoids” across the revised manuscript.

Reviewer

What exactly is shown in Fig. 2? The figure legend provides conclusions but not a description of what was done experimentally. If I assume that the image represents SEM of cells following exposure to neodymium, then in order to be able to reach the stated conclusion that cell shape and retention of liquid is preserved under a low vacuum, I would need to see an image captured in the same way of cells prepared in parallel but not exposed to neodymium, and cells prepared in parallel but not exposed to vacuum. These important controls are missing here and must be provided.

Response

Thank you very much for pointing this discrepancy. We changed the Fig. 2 description. Regarding SEM images and cell shape preservation, detailed information on the retention of cell volume after exposure to neodymium solution and under control conditions is referring to our previous work [Novikov, I., Subbot, A., Turenok, A., Mayanskiy, N. & Chebotar, I. A rapid method of whole-cell sample preparation for scanning electron microscopy using neodymium chloride. Micron 124, 102687 (2019).]. It is not possible to obtain an image in the same way as shown in Figure 2 without subjecting the sample to vacuum. At the same time,  according to your recommendation we made a control image without neodymium processing, it is shown in the. Fig.2. The brightness of cell structures on control images is much lower than that of crystalline, salted out of the solution upon drying.

“Figure 2. SEM image (in BSD mode) of control cells - without treatment (A) and BR20min - cells after 20min of lanthanoid treatment (B), followed by water washing, 10 min at vacuum (70 Pa). Scale bar: 10 µm.””

Reviewer

I was not able to locate or view the videos, since I could not find a way to access the Supplementary Information. I imagine that these data would be quite valuable.

Response

Thank you very much for pointing this technical miss. We corrected it and uploaded the video both to the journal website and also secured an alternative location for video files. The direct links are:

https://bioree.ru/wp-content/uploads/2020/10/a549_nd_1080.mp4

https://bioree.ru/wp-content/uploads/2020/10/a549_nacl_1080.mp4

Reviewer

Similar to the situation for Fig. 2, in Fig. 3 the figure legend provides conclusions but not a description of what was done. I think that the title of the Figure could be “During exposure to NdCl3, cell movement decreases, while after washing it increases above initial levels”, but the content of the legend itself should describe how cell motility was measured, with how many cells and under which precise conditions, and what the lines on the plot represent (mean? median?) and what the individual points on the plot represent (individual cells? individual parts of a single cell? a composite value for multiple cells within a single field/frame?). The conclusions should be provided in the body text, not the figure legend.

Response:

We provided a more detailed description of the image processing algorithm (Fig.S1) and changed the Figure 3 descriptions as follows:

”Figure 3. Diagram of changes in the magnitude of cell motility before (dark green), during (magenta), and after (light green) lanthanoid treatment - BR20minRev (A) and in control (B) sample.”

We have also added illustration Fig. S1 Image Processing Algorithm

Reviewer

How was the Brownian motion assessed? Were inanimate objects (beads, killed cells) measured in a parallel set of experiments? This is an important control to establish what indeed represents Brownian motion.

Response

Dynamically averaged data from areas where cell elements were absent at all-time denote “Brownian motion” level. We pointed this in the image processing algorithm in the newly added explanatory illustration. (Fig.S1).

Reviewer

It is stated (line 201) that “signs of cell death” were not observed. What kinds of observations would have been taken as a sign of cell death? Are the assays employed capable of detecting cell death? The control experiment suggested above (treating cells in a way as to kill them, to measure what true Brownian motion would look like) should also be performed here to show the reader what death would look like, with regard to morphology and MitoTracker signal as assessed by microscopy. I, for one, do not know what MitoTracker signal would look like in a dead cell of this type.

Indeed, for each of these experimental approaches, it would be important to have a control experiment in which cells are treated in a way that should induce irreversible damage/death so that the reader knows what the effects of such treatment would look like in the assays employed here, and how they compare to what is observed.

Response

We thank the Reviewer to point this important issue. Indeed, we did not use the term “signs of cell death” properly in this part of the text.  We paraphrased the sentence, replacing the phrase “signs of death” with the term «gross disturbances of cell structures». The presence of the Mitotracker signal in the image 4 series only confirms the preservation of the mitochondrial membrane integrity after treatment with lanthanides. We also added a new paragraph with the estimation of “vitality”, to evaluate the permeability of the membranes for DAPI and mitotracker distribution.

The changes in the text:

We did not observe any alteration of cell morphology after lanthanoid treatment – there were no gross disturbances of cell structures, the shape of the cells remained intact during exposition time and right after washing fluorescent labeling of mitochondria could be maintained throughout the procedure, which confirms the preservation of the mitochondrial membrane integrity (Fig.4).

3.2. Preservation of organelles in cells after recovery from the lanthanoid-induced state.

The number of cells with the damaged membrane (and thus permeable for trypan blue) was 7.86±0.36 after 1hour NdCl3 treatment and replenishing the ions concentration (BR1hRev group). After 24 hours of restoration, this value was 5.22±0.55.

For cells treated with NdCl3 for 4h this index significantly reduced after 24 hours of restoration– from 53.1±2.13% to 28.67±1.95% – indicating that cells had recovered.

The number of trypan-positive cells in control vessels was unchanged and amounted to 1.88±0.28 at 0h, and 1.22±0.21 at 24 h.

We perform a control series with ethanol treatment (as known toxicant) and mitotracker staining to illustrate the presence of «live pattern» of distribution after lanthanoid treatment. (Fig.S2 a,b,c)

We also brought results on the permeability for DAPI dye. Data are shown in comparison with control, dead control, and relative toxicant control.

It can be concluded that the effect of lanthanoid treatment does not lead to an increase in such mortality indicator as an increase in membrane permeability and does not change the pattern of mitotracker accumulation for up to 4 hours.””

Reviewer

In lines 217-218 it is stated that the treated cells “divided with the same efficiency as control and doubled their number after 24 hours.” It would be important to see these data as values or as a plot. What does “efficiency” mean, in this context? That the same number of cells divided after 24 hours? Or that the fraction of cells that divided did so at the same rate (speed)? Were the measurements that led to this conclusion performed only at a 24-hr time point? This is how I interpret the description of the experiment in the Methods section (line 139). If so, additional time points in between 0 and 24 hours would be necessary to make conclusions about the “speed” of proliferation, which is an interesting question.

Response

Following the reviewer’s recommendations, we have performed a more time-fractional analysis of cell proliferation after treatment with neodymium. (Fig.S3)

Following text modification was done:

“”3.3. The proliferation of cells after lanthanoid treatment

Knowing that a short treatment with lanthanoids does not lead to visible damage or major cytotoxic consequences for the cells, we further analyzed whether it would have a long-lasting effect on cell biology. We have re-initiated A-549 cells to normal activity level after lanthanoid treatment and found that the part of cells could successfully restore proliferation after four hours in the REEbernation state (Fig.5 B).

Within the first 24 hours after re-activation of the cells incubated for 4 hours with lanthanoids, the space occupied by the cells increased from 19.5% to 25.5%. Remarkably, the main increase of the cells was observed during the first 180 min after reactivation of the cells. After the next  48 hours, there was active growth of the cells, followed by returning of the usual cell dynamics around  72 hours after reactivation from REEbernation state.

Please, note, that in the analysis of the space occupied by the cells, dead cells are also counted, if they adhered to the plastics. Such cells would mask the proliferation dy=uring the first 24-48 hours after re-activation. This is why there is an initial  “plato” on the graphs, showing the dynamic of cell growth.

The shorter (in time) lanthanoid treatment did not change the post-reactivation dynamic of normal growth of cell populations. (Fig. 5A)””

Reviewer

In section 3.3, it is stated that “The number of cells with the damaged membrane (and thus permeable for trypan blue) was significantly reduced” and the values of 53% and 29% are given. To help the reader assess the significance of these values, please provide the raw numbers of cells used to calculate these percentages, and whether these experiments were performed with technical and/or biological replicates. Also, how does 53% of cells with damaged membranes compare to the effects of mock treatments (NaCl) or treatments intended to kill cells? Does having 53% of cells trypan-blue-positive represent massive damage, or is this kind of value consistent with other kinds of treatments that may induce some damage that cells are easily able to recover from? The same question applies to 29% trypan-blue-positive: is this value indicative of persistent damage? What percentage of trypan-blue-positive cells would one expect for an untreated or mock-treated culture?

Response

Thank you very much for pointing this lack of details. We added data regarding the trypan blue staining in the text as follows:

“3.2. Preservation of organelles in cells after recovery from the lanthanoid-induced state.

The number of cells with the damaged membrane (and thus permeable for trypan blue) was 7.86±0.36 after 1hour NdCl3 treatment and replenishing the ions concentration (BR1hRev group). After 24 hours of restoration, this value was 5.22±0.55.

For cells treated with NdCl3 for 4h this index significantly reduced after 24 hours of restoration– from 53.1±2.13% to 28.67±1.95% – indicating that cells had recovered.

The number of trypan-positive cells in control vessels was unchanged and amounted to 1.88±0.28 at 0h, and 1.22±0.21 at 24 h.”

Reviewer

The information provided in Fig. 1 is really only important for interpreting the data in Fig. 5, since the other figures don’t use the same abbreviations to refer to the different treatment and recovery regimens. I therefore suggest that the information in Fig. 1 be moved to a part of Fig. 5, so that the reader can look at them while examining the changes in gene expression.

Response

Thank you very much, for pointing this important issue. We first unified all abbreviations, so that a reader would clearly see the link among experiments and results.

At the same time, we humbly ask the Reviewer to allow us to preserve the scheme of experiments as a Fig. 1, since it is not only reflecting the gene expression sampling design, but also all other experiments, including cell viability, proliferation assay, etc. Thus, as a compromising solution, we place it before reporting the results.

Reviewer

In Fig. 5, is fold change calculated for the first dataset listed compared to the second one? For example, does a logFC of 3 for “BR4hRev vs control” mean that a particular gene was expressed at much higher levels in the BR4hRev sample compared to the control sample? I suspect that this is the case, but I suggest that the authors include some wording to make that point very clear. This is an important point because the authors say that the changes illustrate “restoration of transcriptomic machinery activity”. If the major effect is a resumption of new transcription, then one might expect the majority of genes to have positive logFC values, but that does not appear to be the case here. In general, I think the authors should comment on the relative direction of the changes and how they interpret this.

Response

We thank the Reviewer for pointing the lack of proper description and the explanation. Accordingly, we changed the Figure 5 description.

The Reviewer is absolutely right -, in the whole set of genes with statistically significant changes of expression between BR4hRev and control samples, there are more down-regulated ones. But if we will appreciate only genes having notable expression changes (logFC > |1|), there will be more up-regulated than down-regulated genes (69 vs 24).

“”Figure 5. Changes in gene expression in the cells with neodymium (А, B), RNA-later (E, F), and after the restoration of the cells to the growth medium (C, D). Each point represents a gene with its average level of expression (logCPM, where CPM is “Counts Per Million reads”) and the degree of expression change between given pairs of samples (logFC, where FC is “Fold Change”). The specific logFC rate indicates changes of gene expression in the first sample in pair compared to the second one. Red points correspond to genes with statistically significant changes of expression.”

Reviewer

In the Discussion, the authors say that “RNA transcription is rapidly blocked by NdCl3 treatment”. However, if there are few changes in transcript levels, then mRNA degradation is also blocked, which is worth noting.

Response

Thank you very much, yes, we also tend to think in this way, but specifically for the current manuscript, we are afraid that it is too early to make a statement about the relation REEbarnation and RNAses.

On the other hand, we also know that there is a feature of some lanthanoids to fragment nucleic acids in the non-enzymatic manner (our unpublished observation). At the same time high RIN values, indeed, show that there is no RNA degradation at least for the initial hours. To close this gap, in the current manuscript we added these data to the Supplementary Materials (Fig. S4).

Reviewer

Finally, the changes in expression profile for RNAlater are stated as being “marked” and the data in the table in the SI supposedly supports this statement, but it would be extremely valuable to have those RNAlater data also shown in Fig.5 in the same kind of format (scatter plots), for comparison.

Response

Following the suggestion, we added RNA-later treatment-related data as plots in Fig. 5.

Reviewer 2 Report

The authors have found a way to show down active metabolism reversibly for few hours using neodymium. The results need to be expanded to explain their findings better. 

  1. Figure 1 is not explained any where in the text.
  2. Authors mention that lanthanoid treatment freezes cell movement and preserves 3D shape. Image panels from movie showing time stamps will be good. This is better than showing single image. 
  3. Cell proliferation and trypan blue staining data should be shown in the manuscript. 
  4. More time points to show what happens after reversing lanthanoid treatment must be included. 

Author Response

Response: We sincerely thank The Reviewer 2 for his valuable comments that allowed us to improve the manuscripts with better explanation and clarification of the meanings. The text of the manuscript was updated according to the suggestions.

Reviewer:

Figure 1 is not explained anywhere in the text.

Response:

Figure 1 shows the general design of the experiments referring to different treatments and samples. We have added the corresponding explanations to the text of the manuscript:

“2.1. General study design

In the current study, we analyzed several regimes of incubation of the cell culture with the isotonic lanthanoid-containing solution.  Figure 1 represents a general scheme of sampling and conducted type of analysis”

Reviewer:

Authors mention that lanthanoid treatment freezes cell movement and preserves 3D shape. Image panels from movie showing time stamps will be good. This is better than showing single image. 

Response:

Generally, the analysis of the magnitude of movement provides information on the deceleration of cell motility, and the preservation of the 3D structure after exposure to lanthanides is described in more detail in our previous article. [Novikov, I., Subbot, A., Turenok, A., Mayanskiy, N. & Chebotar, I. A rapid method of whole-cell sample preparation for scanning electron microscopy using neodymium chloride. Micron 124, 102687 (2019).]

Also, following the Reviewer’s suggestion, we added a movie file, illustrating the freeze of the cell movement followed by restoration of the activity after removing the lanthanoid solution (Supplementary data)

Reviewer:

Cell proliferation and trypan blue staining data should be shown in the manuscript.

Response We expanded the section describing proliferation after exposure to lanthanides and provided more detailed  information on trypan blue staining and added other histochemical stains:

“3.2. Preservation of organelles in cells after recovery from the lanthanoid-induced state.

The number of cells with the damaged membrane (and thus permeable for trypan blue) was 7.86±0.36 after 1hour NdCl3 treatment and replenishing the ions concentration (BR1hRev group). After 24 hours of restoration, this value was 5.22±0.55.

For cells treated with NdCl3 for 4h this index significantly reduced after 24 hours of restoration– from 53.1±2.13% to 28.67±1.95% – indicating that cells had recovered.

The number of trypan-positive cells in control vessels was unchanged and amounted to 1.88±0.28 at 0h, and 1.22±0.21 at 24 h.

We perform a control series with ethanol treatment (as known toxicant) and mitotracker staining to illustrate presence of «live pattern» of distribution after lanthanoid treatment. (Fig.S2 a,b,c)

We also brought results on the permeability for DAPI dye. Data are shown in comparison with control, dead control, and relative toxicant control.

It can be concluded that the effect of lanthanoid treatment does not lead to an increase in such mortality indicator as an increase in membrane permeability and does not change the pattern of mitotracker accumulation for up to 4 hours.”

Reviewer :

More time points to show what happens after reversing lanthanoid treatment must be included.

Response:

We followed the Reviewer’s suggestion and added additional time points after reversing lanthanoid treatment of the cells:

“3.3. Proliferation of cells after lanthanoid treatment

Knowing that a short treatment with lanthanoids does not lead to visible damage or major cytotoxic consequences for the cells, we further analyzed whether it would have a long-lasting effect on cell biology. We have re-initiated A-549 cells to normal activity level after lanthanoid treatment and found that the part of cells could successfully restore proliferation after four hours in the REEbernation state (Fig.5 B).

Within the first 24 hours after re-activation of the cells incubated for 4 hours with lanthanoids, the space occupied by the cells increased from 19.5% to 25.5%. Remarkably, the main increase of the cells was observed during the first 180 min after reactivation of the cells. After the next  48 hours, there was active growth of the cells, followed by returning of the usual cell dynamics around  72 hours after reactivation from REEbernation state.

Please, note, that in the analysis of the space occupied by the cells, dead cells are also counted, if they adhered to the plastics. Such cells would mask the proliferation dy=uring the first 24-48 hours after re-activation. This is why there is an initial  “plato” on the graphs, showing the dynamic of cell growth.

The shorter (in time) lanthanoid treatment did not change the post-reactivation dynamic of normal growth of cell populations. (Fig. 5A)”

Round 2

Reviewer 1 Report

I am satisfied with the authors' responses and the improvements they have made to the manuscript.

Reviewer 2 Report

The authors have addressed my comments and I support publication.